# Blind Video Bit-Depth Expansion

## ABSTRACT

With the rapid development of high-bit-depth display devices, bit-depth expansion (BDE) algorithms that extend low-bit-depth images to high-bit-depth images have received increasing attention. Due to the sensitivity of bit-depth distortions to tiny numerical changes in the least significant bits, the nuanced degradation differences in the training process may lead to varying degradation data distributions, causing the trained models to overfit specific types of degradations. This paper focuses on the problem of blind video BDE, proposing a degradation prediction and embedding framework, and designing a video BDE network based on a recurrent structure and dual-frame alignment fusion. Experimental results demonstrate that the proposed model can outperform some state-of-the-art (SOTA) models in terms of banding artifact removal and color correction, avoiding overfitting to specific degradations and obtaining better generalization ability across multiple datasets. The proposed degradation model and source codes will be open-sourced.

## CCS CONCEPTS

• **Computing methodologies → Computer vision tasks**.

## KEYWORDS

Video bit-depth expansion, false contour removal, quantization and dequantization.

| Train \ Test | Ground Truth | Degradation in RGB | Degradation in YUV | Degradation in YCbCr |
|---|---|---|---|---|
| **Degradation in RGB** | | | | |
| PSNR | | **42.04** | 29.54 | 23.25 |
| SSIM | | **0.9887** | 0.9690 | 0.9261 |
| **Degradation in YUV** | | | | |
| PSNR | | 27.26 | **31.09** | 20.45 |
| SSIM | | 0.9254 | **0.9393** | 0.7914 |
| **Degradation in YCbCr** | | | | |
| PSNR | | 22.27 | 19.75 | **32.13** |
| SSIM | | 0.9386 | 0.9334 | **0.9653** |

**Figure 1: The trained LBDEN model [31] tends to overfit the degradation settings of the training data.**

## 1 INTRODUCTION

With the continuous development of display technology, display devices have expanded from traditional 8-bit to higher bit-depths, such as ultra-high-definition (UHD) TVs and some smartphones supporting 10-bit or even 12-bit display, to meet the human visual system's demand for finer grayscale variations. Therefore, there is a need for better conversion of the low-bit-depth (LBD) image and video resources to adapt to high-bit-depth (HBD) display devices. The bit-depth expansion (BDE) task, which aims at recovering high-quality HBD images/videos from LBD ones, has gradually attracted more attention from academia and industry.

Traditional BDE methods include zero-padding (ZP), directly filling missing bits with zeros; multiplication-by-ideal-gain (MIG), commonly multiplying by a gain factor $(2^h - 1)/(2^l - 1)$ where $h$ and $l$ represent high and low bit depths respectively; and bit replication (BR), filling missing bits by replicating known higher bits. These methods can expand bit-depth through simple strategies but often introduce visual artifacts such as flat area banding (false contours) and color distortion. Further improvement is achieved by employing dithering or debanding algorithms [1, 3, 8, 15, 20–22, 24] to alleviate visual artifacts.

In recent years, deep neural networks (DNNs) have brought a significant revolution in the field of image enhancement and restoration. BDE is a typical ill-posed image regression problem, which is highly suitable for being addressed using deep models. Early DNN-based methods introduce classic architectures like UNet and ResNet to remove the flat area banding artifacts [2][11] or predict missing bits [12]. Subsequently, numerous related improvements emerged. For example, Zhao *et al.* [31] propose a lightweight residual-block-in-residual-block structure with dilated convolutions. Punnappurath *et al.* [19] employ a smart strategy of predicting each bit-plane bit-by-bit to enhance prediction accuracy. Liu *et al.* have designed many effective deep models for image BDE task, such as enhanced variational autoencoder model [12], shuffle-based multi-scale fusion network [13] and iteratively recovered residual features [18]. Recently, some video BDE models [14][10] have extended traditional single-image BDE tasks to multi-frame BDE, which can eliminate severe banding effects by utilizing multi-frame information.

Currently, in BDE research, HBD ground truth (GT) images are often degraded to obtain corresponding LBD inputs. Unlike other image restoration tasks, the least significant bits are susceptible to tiny value changes, and the choice of different degradation strategies significantly affects the BDE models. We observe that models trained under different degradation conditions often tend to overfit the specific degradations, thereby reducing the generalization and

| | 1 | 2 | 3 | 4 | 5 | 6 | 7 |
|---|---|---|---|---|---|---|---|
| Color Space | YUV | YUV | YUV | YUV | RGB | RGB | YCbCr |
| $f_Q^\downarrow$ | Ceil;GF1 | Floor;GF1 | Floor;GF2 | Floor;GF2 | Floor;GF1 | Floor;GF2 | Ceil;GF1 |
| $f_{DQ}^\uparrow$ | Ceil;MIG | Floor;MIG | BR | ZP | Floor;MIG | ZP | Ceil;MIG |
| | 8 | 9 | 10 | 11 | 12 | 13 | 14 |
| Color Space | YCbCr | YCbCr | YCbCr | YUV | YUV | YCbCr | YCbCr |
| $f_Q^\downarrow$ | Floor;GF1 | Floor;GF2 | Floor;GF2 | Ceil;GF1 | Floor;GF1 | Ceil;GF1 | Floor;GF1 |
| $f_{DQ}^\uparrow$ | Floor;MIG | BR | ZP | ZP | ZP | ZP | ZP |

**Figure 2: The same image undergoes degradation from 14 different types. Tiny differences in the degradation settings may lead to significant differences in the degraded results.**

robustness of the BDE algorithm in practical applications. As shown in Fig. 1, taking the light bit-depth expansion network (LBDEN) [31] as an example, we employ quantization degradation to obtain 8-bit-to-4-bit BDE training data under different color spaces, i.e., RGB, YUV and YCbCr. The results indicate that these degradations lead to significant banding effects, but the trained LBDEN model is only effective for the banding artifacts consistent with the training scenarios, performing poorly on images with other degradation types. This shows that the current BDE learning methods overfit to specific types of degradation, resulting in poor generalization to other types of data distributions. Similarly, other settings in the degradation process also cause overfitting problems, including different gain factors $2^{h-l}$ and $(2^h-1)/(2^l-1)$, and whether to round down, round up, or round to the nearest integer after dividing by the gain factor.

In practice, we aim for trained BDE models to be effective across various types of data distributions, rather than being limited to specific ones. Therefore, this paper proposes a robust blind video bit-depth expansion (BVBDE) method. Specifically, the proposed method first designs a series of bit-depth degradation models by combining different color spaces, bit-depth quantization strategies, and numerical rounding methods, and then a blend of various degradation types is used to train the BDE model. For a LBD image with unknown data distributions, a pre-trained discriminative model is employed to predict the degradation type that is more similar to the data distribution of the degradation model. The predicted degradation information is then embedded into the BDE network using positional encoding. Additionally, this paper proposes a recurrent dual-frame fusion enhancement network that efficiently utilizes temporal information and neighbor frame details to improve the visual quality of BDE results.

The main contributions of this paper are summarized as follows.

- This paper experimentally points out the significant impact of different degradations on the generalization and robustness of BDE models. Consequently, the task of blind video bit-depth expansion is introduced to avoid overfitting to training data distributions of specific degradation.
- A video bit-depth expansion architecture is designed based on degradation distribution prediction and embedding, which effectively reduces the difficulty of learning multiple degradations simultaneously while enhancing the robustness of the model. In addition, the proposed network makes full use of long-term temporal information and neighbor frame details while balancing the benefits of both sliding window structure and recurrent structure.
- Experimental results demonstrate that the proposed method effectively improves the robustness of BDE models, outperforming some SOTA algorithms on blind BDE scenarios, and achieving better subjective performance in false contouring removal and color correction.

## 2 RELATED WORKS

With the rise of deep learning, it has gradually become the mainstream tool for BDE tasks. Liu *et al.* [11] firstly adopt DNN to restore HBD images. Zhao *et al.* [32] introduce a dual-branch residual BDE network, which removes banding artifacts on flat regions and predicts missing bits for texture regions, respectively. Then, by focusing on flat regions, Zhao *et al.* propose a lightweight and efficient model LBDEN [31] using a residual-block-in-residual-block structure with dilated convolutions. Wen *et al.* [26] employ Transformer blocks to extract multi-scale information, cyclically fusing local information guided by global information. Unlike previous residual architectures, BitNet [2] employs an encoder-decoder architecture with dilated convolutions and multi-scale feature integration.

Instead of directly learning regression from LBD to HBD images, some methods reconstruct residual components through deep

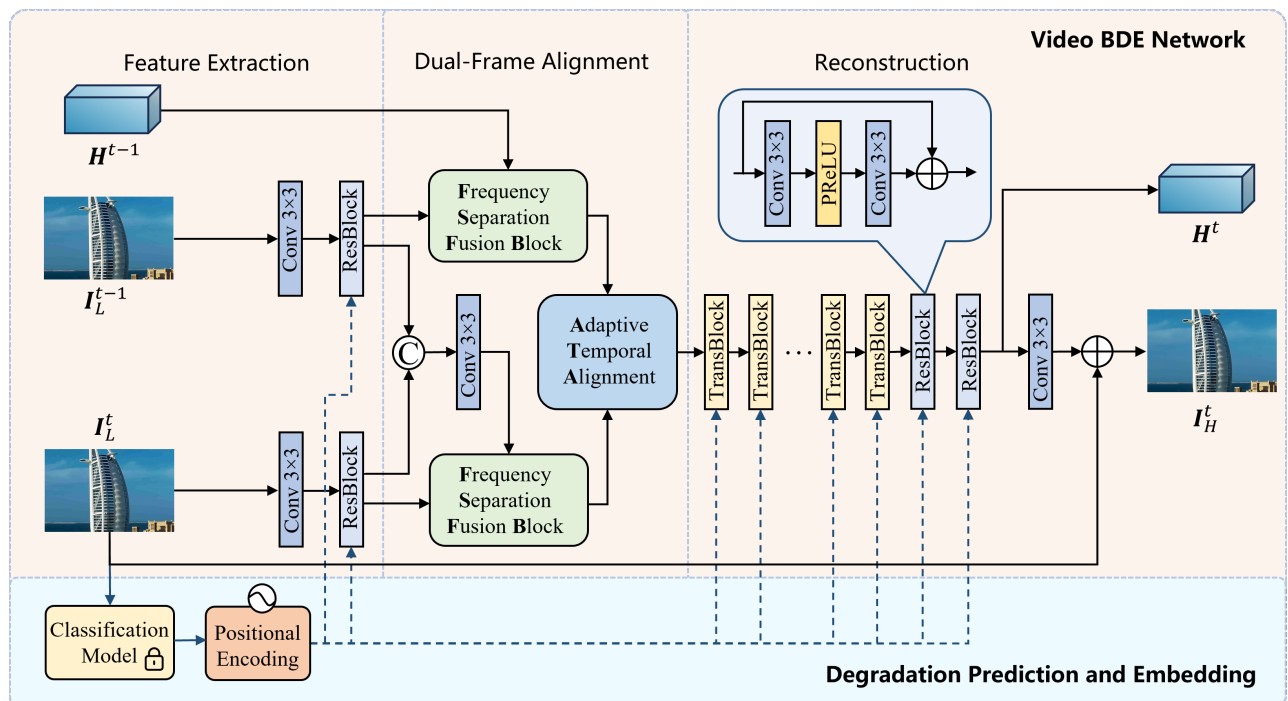

**Figure 3: Framework of the proposed blind video bit-depth expansion (BVBDE) method.**

models. In [19], Punnappurath *et al.* recover each bit-wise map iteratively, achieving impressive performance but requiring training many times. Subsequently, Liu *et al.* [18] decompose the residual maps into different frequency sums, proposing an iterative residual feature optimization strategy. BRNet [16] utilizes network learning to assign weights to each pixel, enabling the restoration of images with different quantization levels. Han *et al.* [5] utilize implicit neural representation and a phase estimator to achieve the recovery of dequantized images from inputs with arbitrary quantization levels.

Recently, some video BDE models have been proposed to leverage multi-frame information. Liu *et al.* [14] present an attention mechanism for implicit alignment to remove banding artifacts across multiple frames. However, due to the high computational complexity of alignment operations, Liu *et al.* [10] subsequently propose a two-stage progressive network for adaptive inter-frame information fusion. Although these models have achieved superior performance by using multi-frame information, their performance on blindly quantized images is still not ideal. Note that Zhao *et al.* [30] propose a blind image de-contouring algorithm, attempting to simultaneously address banding artifacts caused by different degradations such as BDE, image and video coding, and color adjustments. In this paper, we further observe that even different bit-depth degradations will lead to significant generalization issues, so we mainly focus on the blind BDE problem.

## 3 PROPOSED METHOD

### 3.1 Degradation Formulation for Blind BDE

For a $h$-bit HBD image $I_h$ and corresponding $l$-bit LBD image $I_l$, the degradation model in this paper is defined as,

$$I_l = \left[ f_{DQ}^{\uparrow} \left( \left[ f_Q^{\downarrow} \left( f_C \left( I_h \right) \right) \right] \right) \right] \tag{1}$$

where $f_C$, $f_Q^{\downarrow}$, and $f_{DQ}^{\uparrow}$ denote color space conversion, quantization process, and dequantization operation, respectively, and $[\cdot]$ represents a rounding function. Three different rounding functions are used in our experiments, i.e., round down (*floor*), round up (*ceiling*), or round to the nearest integer (*round*). In addition to *RGB* color space, *YUV* and *YCbCr* color spaces are adopted, which are often used in image and video coding. For quantization degradation $f_Q^{\downarrow}$, two types of ideal gain factors (GF) are used, as follows,

$$I_l^* = f_Q^{\downarrow} \left( I_h \right) = \begin{cases} I_h * \frac{2^l - 1}{2^h - 1}, & (GF1) \\ I_h * \frac{1}{2^{h-l}}, & (GF2) \end{cases} \tag{2}$$

where *GF1* factor is commonly used in the multiplication-by-ideal-gain (MIG) algorithm, and *GF2* factor is equivalent to a bitwise right-shift by ($h$-$l$) bits. These two factors are also employed during the dequantization stage. Multiplying by *GF1* factor is the MIG while multiplying by *GF2* factor corresponds to zero padding (ZP). Additionally, traditional bit replication (BR) methods are also utilized in the dequantization process, as follows:

$$I_h^* = f_{DQ}^{\uparrow} \left( I_l^* \right) = \begin{cases} I_l^* * \frac{2^h - 1}{2^l - 1}, & (MIG) \\ I_l^* * 2^{h-l}, & (ZP) \\ \text{shift}_L^{(h-l)} \left( I_l^* \right) + \text{bitcopy} \left( I_l^* \right), & (BR) \end{cases} \tag{3}$$

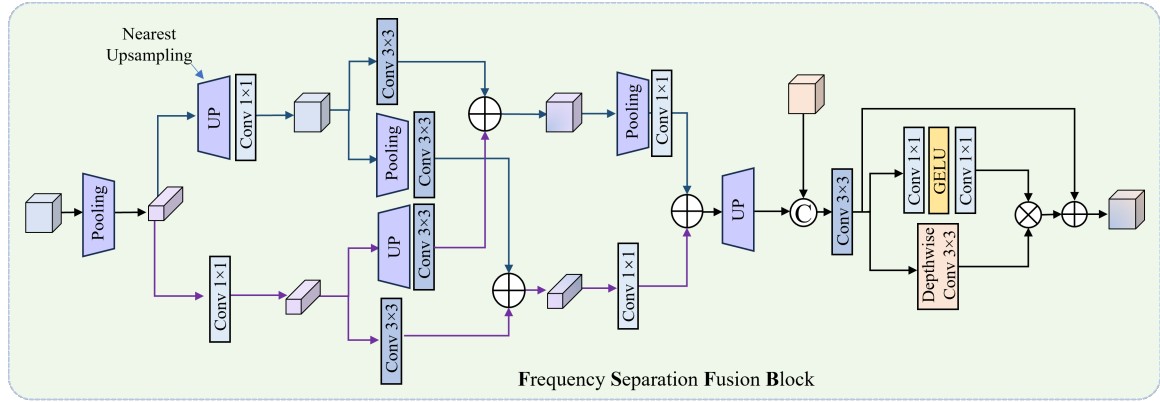

Figure 4: Structure of frequency-separated fusion block (FSFB).

where $shift_L$ denotes bitwise left-shift, and $bitcopy$ represents bit replication operation that fills unknown $(h - l)$ bits using known bits.

Theoretically, this degradation model contains a total of 54 (3 × 3×2×3) different degradation combinations. However, some combinations may result in unreasonable and erroneous values in practice. For example, using the $GF2$ factor, $ZP$ factor, and ceiling operations in the $YUV$ color space may produce artifacts such as black dots. We further filter out degradation combinations that lead to visual artifacts and then merge types of degradation with the same numerical distributions, ultimately obtaining 14 different degradation scenarios, as illustrated in Fig. 2. We can observe that the BDE degradation model exhibits high sensitivity to numerical values. Taking the first two classes as examples, even different rounding functions may cause significantly different color deviations and banding artifacts.

## 3.2 The Framework of Blind Video BDE Method

The framework of the proposed BVBDE method is illustrated in Fig. 3, which consists of a recurrent dual-frame enhancement module and a degradation distribution prediction and embedding module. Multi-frame enhancement networks often employ sliding-window-based architectures or recurrent neural network (RNN) structures. The sliding-window architecture can fully utilize spatial details of neighbor frames by aligning and fusing multiple frames within the window. However, its drawback lies in its inability to capture long-term temporal information, and high computational cost for multi-frame alignment and fusion. The RNN structure can transmit long-term temporal information through hidden features, and the computational complexity of each recurrent cell is close to that of a single-frame model. However, RNN-based structure cannot fully exploit the spatial details of neighbor frames. Therefore, by balancing the advantages of these two types of multi-frame enhancement architectures, the proposed BVBDE adopts a recurrent structure, and fuses two adjacent frames in each recurrent cell, as follows,

$$I_H^t, H^t = F_{BVBDE} \left( I_L^{t-1}, I_L^t, H^{t-1}, c, \theta \right) \quad (4)$$

where $I_L^t$ and $I_H^t$ denote LBD input and reconstructed HBD result of the $t$-th frame, $H^t$ represents the hidden state features of the

$t$-th frame, $F_{BVBDE}$ denotes the BVBDE network with learnable parameters $\theta$. The symbol $c$ represents the embedded information of the degradation distribution predicted by a pre-trained classification model $F_D$, calculated as follows:

$$c = PE \left( F_D \left( I_L^t \right) \right) \quad (5)$$

where $PE(\cdot)$ denotes the positional encoding operation as in Transformer network [23]. It is worth noting that frames of one video sequence typically have the same bit-depth degradation. Therefore, for each video sequence, the discriminator network $F_D$ runs only once, and the same encoded distortion prediction information $c$ is embedded into the calculation of each frame.

## 3.3 Details of Each Module

As shown in Fig. 3, the BVBDE network comprises a shallow feature extraction module, a dual-frame alignment module, a reconstruction module, and a degradation embedding module.

**Shallow feature extraction module.** This input module consists of a 3×3 convolutional layer and a normal residual block (ResBlock) [6]. In each recurrent cell, current frame $I_L^t$ and the previous neighbor frame $I_L^{t-1}$ both independently utilize the feature extraction module.

**Dual-frame alignment module.** Banding or false contour artifacts often occur in low-frequency flat regions. Therefore, we tend to enforce the network to differentiate between flat regions and high-frequency texture regions during optimization. Motivated by the frequency-separation deblurring method [29], pooling operations can be used to downsample features to compel the model to separately process the downsampled low-frequency features and the high-frequency features. Thus, a frequency-separated fusion block (FSFB) unit is designed, as illustrated in Fig. 4. First, the pooling layer is employed to increase the receptive field and reduce computational complexity, which also discards some high-frequency details to focus on the low-frequency regions where banding artifacts are more significant. Subsequently, two branches are simultaneously utilized to process features in downsampling and upsampling spaces, respectively, to perceive the differences between high and low-frequency features. Finally, the high and low-frequency features are added together and upsampled, and

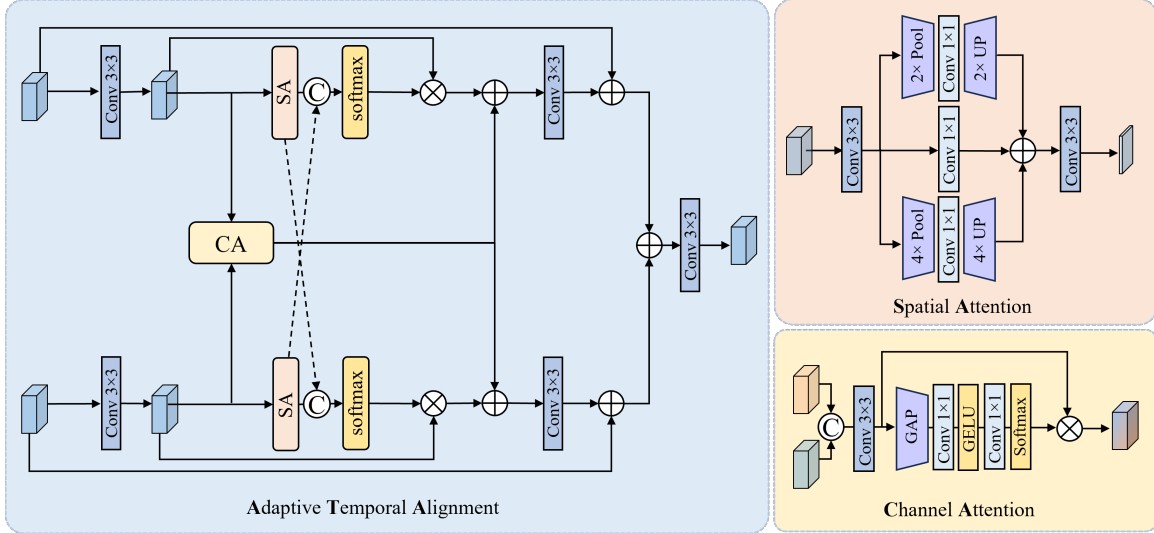

Figure 5: Structure of adaptive temporal alignment unit (ATA).

then fused through a simple feature fusion process. In the proposed method, one FSFB unit is used to fuse the relevant features between the previous frame $I_L^{t-1}$ and previous hidden features $H^{t-1}$, the other FSFB unit fuses the features between the current frame $I_L^t$ and the concatenated frames. These two FSFB units share weights.

After FSFB calculation, historical features and current-frame features undergo alignment fusion through an adaptive temporal alignment unit (ATA). Fig. 5 illustrates the structure of ATA, which primarily employs cross-branch merging to fuse features and utilizes channel and spatial attention mechanism to improve fusion effectiveness. Commonly used channel attention (CA) and spatial attention (SA) structures are used in this unit with minor modifications. A concatenation operation is introduced in the CA to capture interactions between historical and current features. Since the banding artifacts often occur over large areas in flat regions, multi-branch and multi-scale convolutions are adopted in the SA to obtain a larger receptive field. Both CA and SA apply the Softmax function to normalize the pixels at each position along the channel direction, enabling the network to focus on relevant channels.

**Reconstruction module.** The reconstruction stage of this method consists of 10 Transformer blocks and 2 residual blocks. To reduce the computational cost, lightweight Transformer blocks (TransBlock) [28] [9] are applied, which reduces the dimensionality of the computed attention maps from $\mathbb{R}^{HW \times HW}$ to $\mathbb{R}^{C \times C}$ through dimension exchange, allowing direct application to compute image features without patch-level tokenization. We set the number of attention heads in these blocks to 1 and add a branch with dilated convolutions in the feed-forward network (FFN) to capture more spatial information. In our experiments, the dilation parameters in these TranBlocks are empirically set to [1, 1, 2, 2, 3, 3, 2, 2, 1, 1]. Details of these TransBlocks can be found in the supplementary materials.

**Degradation embedding module.** For blind LBD input, a pre-trained classification model is used to predict which class of degradation data distribution is more similar to the data distribution of

the input image. In this paper, MobileNetV1 [7] is adopted as the classification model, and is separately trained with different degradation types as class labels (1~14). Visual artifacts of BDE mainly arise from the least significant bits, which are easily overlooked by the classification model due to their small values. Therefore, during the training of the classification model, the least significant $l$ bits are extracted by bitwise left-shift and then concatenated with the original image to input into the classification model, allowing the model to focus on the least significant bits. As mentioned before, this model is computed only once for an entire video sequence.

Inspired by the positional encoding in Transformer models [23], the degradation distribution prediction result of the classification model is also embedded into the BDE network through positional encoding operation. After positional encoding, a 4-layer multi-layer perceptron (MLP) with ReLU activation function is employed. The dimension of the encoded result is $\mathbb{R}^{N \times C \times 1 \times 1}$, which is then added to the features of each block.

## 3.4 Loss functions

The proposed BVBDE network is trained by constraining the similarity between the output $I_H^t$ and the ground truth $O_H^t$. For each training video sequence with $N_T$ frames, we use common L1 loss $\mathcal{L}_1$ and SSIM loss $\mathcal{L}_{ssim}$ to optimize the BDE model, as following:

$$\mathcal{L} = \frac{1}{N_T - 1} \sum_{t=2}^{N_T} \left( \mathcal{L}_1 \left( I_H^t, O_H^t \right) + \mathcal{L}_{ssim} \left( I_H^t, O_H^t \right) \right) \qquad (6)$$

## 4 EXPERIMENT

### 4.1 Datasets and Implementational Details

**Blind Video BDE Dataset.** To build the dataset for blind video BDE, many 8-bit 4K (3840 × 2160) videos are collected from the Internet and further downsampled to 2K resolution (1920 × 1080). These videos are then randomly degraded for 8-bit-to-4-bit and 8-bit-to-6-bit scenarios through the aforementioned 14 classes of

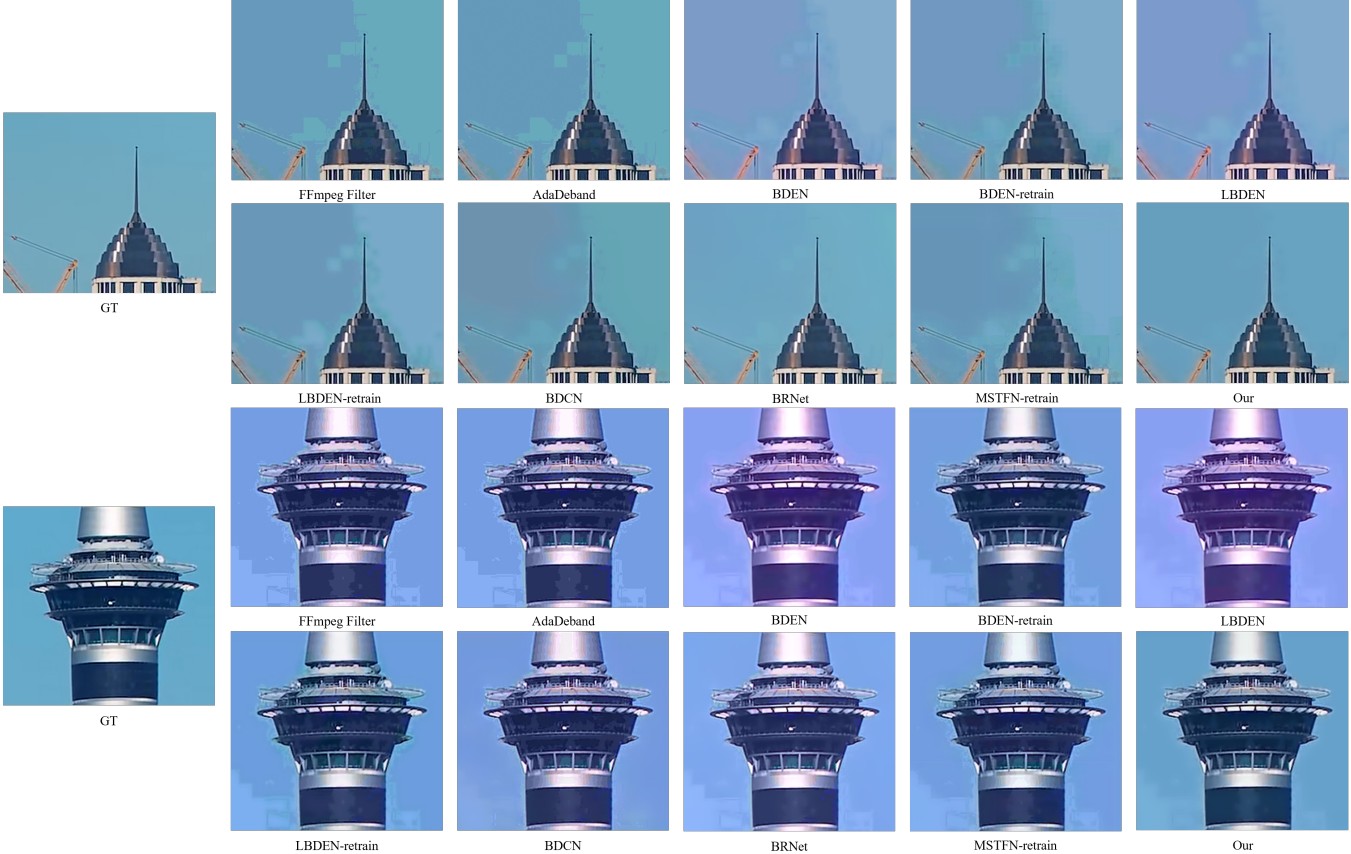

**Figure 6: Visualizing the restoration of a single frame of a 4-8-bit video using various BDE algorithms.**

**Table 1: The quantitative results of 4-8-bit and 6-8-bit on the blind video BDE dataset. The best result is marked in bold, and the second-best result is underlined.**

|  |  | FFmpeg Filter | AdaDeband | BDEN | BDEN-retrain | LBDEN | LBDEN-retrain | BDCN | BRNet | MSTFN | Our |
|---|---|---|---|---|---|---|---|---|---|---|---|
| 4bit | PSNR↑ | 27.69 | 27.66 | 26.07 | 29.43 | 25.75 | 29.12 | 28.12 | 28.22 | 29.00 | **31.30** |
|  | SSIM↑ | 0.9380 | 0.9392 | 0.9730 | 0.9718 | 0.9733 | 0.9695 | 0.9618 | 0.9745 | 0.9591 | **0.9804** |
|  | LPIPS↓ | 0.2888 | 0.2897 | 0.1406 | 0.1510 | 0.1472 | 0.1717 | 0.1516 | 0.1462 | 0.1912 | **0.0791** |
|  | Time(s) | / | / | / | 0.1279 | / | 0.0499 | **0.0193** | 0.1008 | 1.0200 | 0.0371 |
| 6bit | PSNR↑ | 38.58 | 38.81 | 37.19 | 40.01 | 36.97 | 39.76 | 37.02 | 39.36 | 39.09 | **41.65** |
|  | SSIM↑ | 0.9841 | 0.9878 | 0.9939 | 0.9942 | 0.9941 | 0.9942 | 0.9933 | 0.9931 | 0.9906 | **0.9953** |
|  | LPIPS↓ | 0.0595 | 0.0718 | 0.0292 | 0.0291 | 0.0260 | 0.0319 | 0.0294 | 0.0480 | 0.0582 | **0.0161** |
|  | Time(s) | / | / | / | 0.1281 | / | 0.0493 | **0.0192** | 0.0992 | 1.0200 | 0.0361 |
|  | Parameters(k) | / | / | / | 2100 | / | **183** | 1232 | 4086 | 4227 | 214 |

quantization degradation methods. Finally, the training set contains 428 video sequences with 50 frames for each sequence, and the test set consists of 97 sequences with 5 frames for each sequence.

**Generalization Test Sets.** To evaluate the generalization ability of BDE models, we add two public video datasets of REDS [17] and Vimeo-90K [27], and another real-world test set. For the REDS test set, the first 107 sequences from the training set are selected for testing. For the Vimeo-90K set, we select one sequence from each folder, resulting in a total of 96 sequences. The degradation types

of these test sequences are also randomly selected. To create the real-world test set, we collect 9 sequences with unknown banding artifacts from the Internet, each sequence contains 5 frames.

**Implementational Details.** To facilitate the learning of historical information calculation and updating during the training of the recurrent network, three recurrent cells are concatenated in series and optimized together, with each cell outputting one reconstructed frame. The training images are further cropped into image patches with 128×128 resolution. We utilize the Adam optimizer and employ

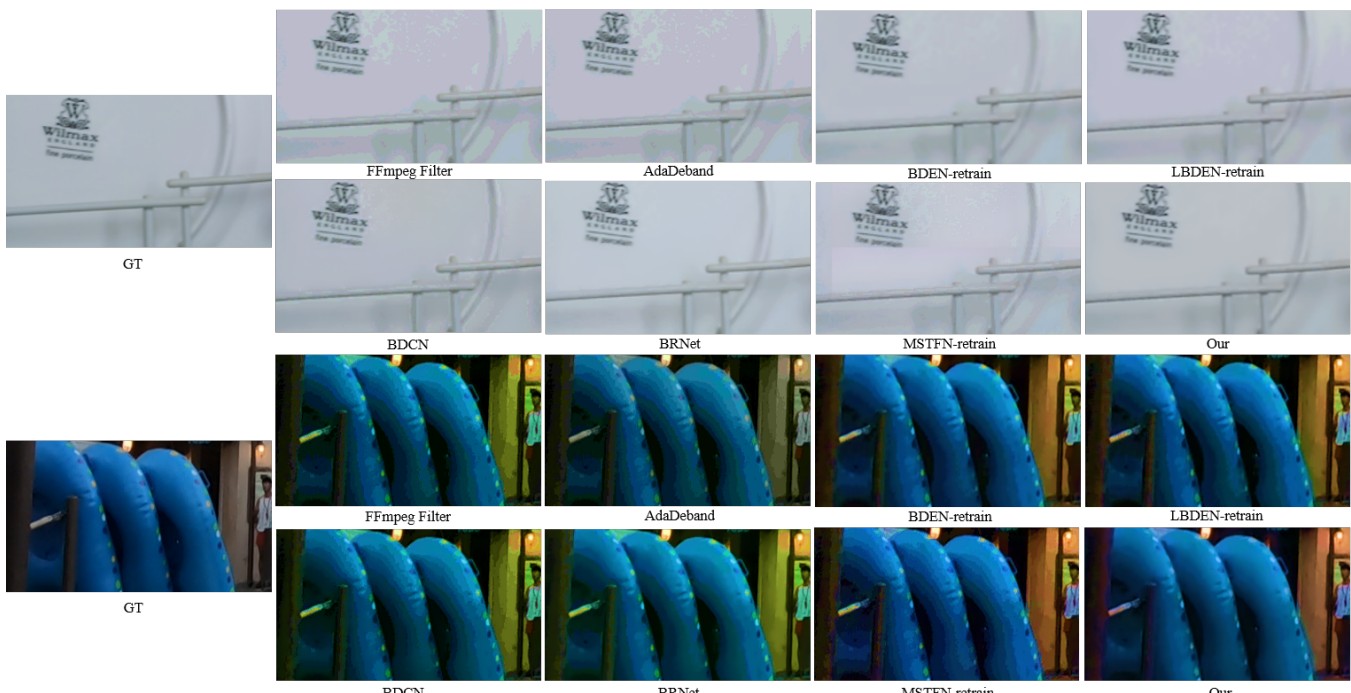

**Figure 7: Visualization of 4-8-bit results on the Vimeo-90K and REDS datasets. The top section shows results on the Vimeo-90K dataset, while the bottom section displays results on the REDS dataset.**

**Table 2: The quantitative results of different methods on the REDS and Vimeo-90K test sets.**

| | | | FFmpeg Filter | AdaDeband | BDEN-retrain | LBDEN-retrain | BDCN | BRNet | MSTFN | Our |
|---|---|---|---|---|---|---|---|---|---|---|
| REDS | 4-8-bit | PSNR↑ | 27.87 | 27.89 | 29.96 | 29.76 | 28.14 | 28.85 | 29.37 | **31.49** |
| | | SSIM↑ | 0.9199 | 0.9219 | 0.9493 | 0.9454 | 0.9319 | 0.9530 | 0.9365 | **0.9564** |
| | | LPIPS↓ | 0.1373 | 0.1373 | 0.0961 | 0.1021 | 0.1110 | 0.0958 | 0.0987 | **0.0769** |
| | 6-8-bit | PSNR↑ | 38.11 | 38.70 | 39.78 | 39.51 | 37.76 | 39.18 | 38.88 | **41.75** |
| | | SSIM↑ | 0.9845 | 0.9905 | 0.9926 | 0.9926 | 0.9916 | 0.9926 | 0.9907 | **0.9934** |
| | | LPIPS↓ | 0.0211 | 0.0200 | 0.0148 | 0.0155 | 0.0206 | 0.0167 | 0.0172 | **0.0109** |
| Vimeo-90K | 4-8-bit | PSNR↑ | 28.12 | 28.09 | 29.64 | 29.32 | 28.73 | 28.47 | 29.14 | **31.40** |
| | | SSIM↑ | 0.9149 | 0.9169 | 0.9471 | 0.9475 | 0.9394 | 0.9530 | 0.9331 | **0.9612** |
| | | LPIPS↓ | 0.1761 | 0.1775 | 0.1011 | 0.1118 | 0.1097 | 0.0852 | 0.1251 | **0.0722** |
| | 6-8-bit | PSNR↑ | 38.66 | 39.06 | 39.82 | 39.64 | 37.58 | 39.65 | 38.80 | **41.60** |
| | | SSIM↑ | 0.9823 | 0.9874 | 0.9894 | 0.9914 | 0.9901 | 0.9913 | 0.9881 | **0.9924** |
| | | LPIPS↓ | 0.0231 | 0.0256 | 0.0145 | 0.0156 | 0.0218 | 0.0147 | 0.0240 | **0.0106** |

a cosine annealing learning rate schedule with a batch size of 4, initializing the learning rate to $4 \times 10^{-4}$. The proposed method is implemented on the PyTorch platform and NVIDIA TITAN V GPU.

## 4.2 Experimental Results

**Quantitative results.** The proposed method is compared with two non-learnable debanding methods of FFmpeg Filter [4] and AdaDeband [8], a blind image debanding network BDCN [30], two image BDE networks of BDEN [32] and LBDEN [31], and a video BDE model MSTFN [10]. For BDCN, we directly use the pre-trained model for testing. Although BRNet does not have publicly available

training codes, the pre-trained BRNet [16] models can support BDE from arbitrary bits to 8 bits. For fair comparison, the BDEN, LBDEN, and MSTFN have also been retrained on the proposed blind video BDE dataset.

The quantitative results of 4-8-bit and 6-8-bit tests are listed in Table 1. Firstly, by comparing the original BDEN and LBDEN with the retrained BDEN and LBDEN, it can be observed that the original BDEN and LBDEN overfit to specific training distributions, resulting in worse performance than directly applying debanding algorithms for blind BDE. However, through retraining on the proposed dataset, these models exhibit significant improvements

**Table 3: Quantitative Results in Ablation Study.**

| | | BVBDE | w/o degradation embedding | embedding degradation GT labels | w/o FSFB | w/o dilated convolution in TransBlock |
|---|---|---|---|---|---|---|
| 4-8-bit | PSNR↑ | 31.30 | 30.96 | 32.43 | 31.16 | 30.92 |
| | SSIM↑ | 0.9804 | 0.9793 | 0.9804 | 0.9799 | 0.9782 |
| | LPIPS↓ | 0.0791 | 0.0886 | 0.0763 | 0.0821 | 0.0998 |
| 6-8-bit | PSNR↑ | 41.65 | 40.83 | 43.00 | 42.08 | 42.15 |
| | SSIM↑ | 0.9953 | 0.9952 | 0.9952 | 0.9951 | 0.9949 |
| | LPIPS↓ | 0.0161 | 0.0175 | 0.0140 | 0.0171 | 0.0186 |

**Table 4: Comparison of different inter-frame alignment methods on 4-8-bit blind video BDE task.**

| | PSNR↑ | SSIM↑ | LPIPS↓ | FLOPs |
|---|---|---|---|---|
| Adaptive temporal alignment | **31.30** | 0.9804 | 0.0791 | 14.38G |
| Concat | 30.89 | 0.9747 | 0.1187 | **3.79G** |
| PCD module | 31.25 | **0.9813** | **0.0787** | 76.12G |

and can handle data distribution differences caused by different degradations. Secondly, the results of the proposed method significantly outperform the retrained SOTA methods BDEN, LBDEN, and MSTFN, which demonstrate the effectiveness of the proposed degradation prediction and embedding framework and the recurrent dual-frame alignment network. Finally, the number of parameters and average runtime for each method are also listed in Table 1. The proposed method can achieve SOTA results with a relatively lightweight structure. Note that for each video sequence, the classification model is only computed once (6 ms), so the additional runtime of the classification model becomes proportionally smaller as the number of frames in the sequence increases.

Table 2 presents the test results of these methods on the REDS and Vimeo-90K test sets. It can be seen that the SOTA BDE models trained on the proposed dataset exhibit good generalization performance when tested on cross-datasets. Furthermore, the proposed method still achieves better scores than other SOTA models in both distortion and perceptual metrics.

**Visual quality comparisons.** Some 4-8-bit results of different methods are illustrated in Fig. 6. It can be observed that the proposed method effectively removes severe banding artifacts and achieves better subjective quality than other SOTA methods. Additionally, other methods may result in varying degrees of color shifts, whereas the proposed method can accurately restore color information through perception and mapping of different data distributions. Fig. 7 shows some 4-8-bit results of different methods on REDS and Vimeo-90K test sets. We can find that the proposed method robustly removes banding artifacts caused by different degradations and restores high-fidelity colors.

In addition, extended experiments are implemented on several real-world videos with banding artifacts. Related visual results and mean opinion scores can be found in the supplementary materials.

## 4.3 Ablation study

Table 3 lists the results of ablation experiments. The proposed BVBDE model perceives the degradation type of the input LBD image through a pre-trained classification model. Therefore, we compare the results with two different embeddings, i.e., one without using any degradation type embedding and the other using ground truth labels. It can be observed that perceiving the degradation type during the training phase effectively improves the final results. Furthermore, if the classification model can be more accurate, there is still room for further improvement in reconstruction quality. However, a larger classification model also means increased computational cost for degradation prediction.

In addition, Table 3 also shows the results of removing the frequency separation fusion block (FSFB) unit and the dilated convolution branch in the feed-forward network of TransBlock. It can be found that these designs can also bring improvements.

Table 4 compares the results of different inter-frame alignment methods. It can be seen that the proposed adaptive temporal alignment (ATA) unit and common pyramid cascade deformable (PCD) module [25] outperform the direct concatenation. The proposed alignment unit can achieve similar performance to the PCD module, but with less than one-fifth of the computational complexity.

## 5 CONCLUSION

This paper analyzed the problem of current bit-depth expansion (BDE) networks overfitting to specific degradations and thus proposed a blind video BDE algorithm. The proposed method synthesizes and merges 14 types of degradations by analyzing the computational differences in the quantization and dequantization processes, pre-trains a classification model to perceive the degradation distribution types of input low bit-depth images, and then embeds degradation prediction into the learning process of the BDE network. An efficient blind video BDE network is proposed based on recurrent structure and dual-frame alignment fusion. In addition, some detailed modules are designed or improved, such as frequency separation fusion block and adaptive temporal alignment. Experimental results demonstrate that the proposed method can outperform SOTA methods in banding artifact removal and color restoration, and achieves better generalization and robustness across multiple datasets.

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
