# OpenReview forum: "Blind Video Bit-Depth Expansion"
_acmmm.org/ACMMM/2024/Conference — MM2024 Poster_

### Official Review · Reviewer_mPjm · 2024-05-11

**Rating:** 2
**Confidence:** 3

**Summary:**

The paper addresses the challenges in bit-depth expansion (BDE) algorithms caused by overfitting to specific degradation types during training. It introduces a blind video BDE approach incorporating a degradation prediction and embedding framework, alongside a video BDE network with recurrent structures and dual-frame alignment fusion. Experimental results demonstrate superior performance in banding artifact removal and color correction, with better generalization across diverse datasets.

**Strengths:**

The visualizations are clear and easy to understand.

**Limitations:**

1. The author mentioned that the degradation distribution prediction is achieved through a classification network. What is the dimensionality of its output? Currently, class labels range from labels 1 to 14. What happens if it exceeds this range? Additionally, the purpose of the positional encoding operation is not very clear. Could the author provide a detailed explanation?
2. The authors claim the generalization ability of different datasets and test on the REDS and Vimeo-90K. The author should explain the difference between these two datasets. The author collected the training data from the Internet. Whether the training data is similar to the test data?  Whether it can work on the cartoon dataset or other special datasets.
3. In Fig.3, there are two inputs for FSFB, but there is only one input in Fig.4. The author should check whether there is a mistake.
4. How to select the testing sequence in Vimeo-90K, what is the selection rule?
5. In line 637, what is the proportion of each distortion type? what are the distortion types in the test sets？
6. The model includes a classification network, but Table1 shows that the model only has 214K. Could you provide more details about model size?
7. The authors should carefully review for punctuation errors and other potential flaws: e.g., the formula lacks punctuation at the end, e.g., eq(6).
8. Suggestion: it would be better to provide the results of the input (PSNR/SSIM/LPIPS) in Table 2.

**Suitability:**

2

---

### Official Review · Reviewer_FUG3 · 2024-05-25

**Rating:** 6
**Confidence:** 4

**Summary:**

This paper proposes a degradation prediction and embedding framework, and designs a video BDE network based on a recurrent structure and dual-frame alignment fusion. Experimental results on several typical datasets demonstrate that the proposed model can outperform some state-of-the-art (SOTA) models in terms of banding artifact removal and color correction.  The paper is well written and organized.

**Strengths:**

1.The novelty is clearly stated. The paper proposed a video bit-depth expansion architecture based on degradation distribution prediction and embedding. In addition, the proposed network makes full use of long-term temporal information and neighbor frame details while balancing the benefits of both sliding window structure and recurrent structure.

2.The experiments are very comprehensive and detailed comparative experiments are conducted for various situations.The experimental results verified the effectiveness of the proposed algorithm.

3.The references cited in this paper are quite comprehensive, and the summary are quite accurate.

**Limitations:**

1. The Blind Video BDE Data is crucial to this paper, and the introduction of this dataset should be explained more specifically.

2.The order of references needs to be adjusted. Some papers are cited at the beginning while appears in the later order, which affects the readability of the paper. For example, some video BDE models [14][10].

3.For the related work, it seems that  Liu has adopted DNN to restore HBD images, "Liu et al. [11] firstly adopt DNN to restore HBD images." For a better understanding, this paper needs to provide a more detailed introduction.

4.For better understand the advantages and disadvantages of various algorithms in the experimental results, it is recommended to add a reference to the corresponding literature number after the corresponding abbreviation.

**Suitability:**

3

---

### Official Review · Reviewer_pzkW · 2024-05-26

**Rating:** 4
**Confidence:** 4

**Summary:**

This paper proposes a robust blind video BDE network , dubbed as BVBDE. The BVBDE network makes use of a pre-trained classification model to predict the degradation types of input LBD images, and then embeds the predicted degradation information into the BVBDE network using positional encoding. The BVBDE network also employs a recurrent dual-frame fusion enhancement module to efficiently utilize both temporal information and neighbor frame details to improve the visual quality of BDE results. Experimental results demonstrate that the proposed BVBDE network can outperform other SOTA BDE methods in banding artifact removal and color restoration, and achieves better generalization and robustness across multiple datasets.

**Strengths:**

For the proposed BVBDE network, its degradation distribution prediction and embedding module can effectively reduce the difficulty of learning multiple degradations simultaneously while enhancing the robustness of the model. Meanwhile, its recurrent dual-frame fusion enhancement module can make full use of long-term temporal information and neighbor frame details while balancing the benefits of both sliding window structure and recurrent structure.

**Limitations:**

Please do not mix 'module' and 'network' casually in this article, for example, "this paper proposes a recurrent dual-frame fusion enhancement network". should be "... a recurrent dual-frame fusion enhancement module", which would easily make confusion.

More current state-of-the-art video BDE methods should be compared with the proposed

Besides MSTFN, the proposed BVBDE should be compared with other state-of-the-art video BDE methods, such as TANet, SSCNN (Spatiotemporal Symmetric Convolutional Neural Network).

(SSCNN)"Spatiotemporal Symmetric Convolutional Neural Network for Video Bit-Depth Enhancement" TMM 2019, should also be cited and discussed in the related works (Sec.2).

The structure of TransBlock in the proposed BVBDE network should be illustrated in a separate figure.

Besides less challenging BDE scenarios such as 4-8bit and 6-8bit, the proposed BVBDE should be evaluated and compared with other compared BDE methods at more challenging BDE scenario such as 4-16bit based several large-scale HBD video datasets, i.e. COSMOS, Sintel and Tears of Steel (TOS).

**Suitability:**

3

---

### Official Review · Reviewer_kiRH · 2024-05-29

**Rating:** 2
**Confidence:** 3

**Summary:**

The paper introduces the blind video bit-depth expansion (BVBDE) method. This method involves the construction of bit-depth degradation models by combining different:

(i) Color spaces
(ii) Bit-depth quantization strategies
(iii) Numerical rounding methods

After construction, LBD frames are generated as a blend of various degradation types which are then used to train the BDE models.

**Strengths:**

(i) Shallow feature extraction module

(ii) Dual-Frame Alignment Block:

Now, the banding and contour artifacts are present in the low-frequency flat regions. This module splits the input features into low-frequency features and high-frequency ones and then concatenates them to produce features having knowledge of the differentiation.

Then the output of both the FSFBs is passed to the Adaptive Temporal Alignment to fuse the features and process the low-frequency regions.
CA is used for fusion and SA is used for removing the artifacts.

**Since both the current and previous frames are used, the model captures temporal information.

(iii) Reconstruction Module: This part along with the Degradation Embedding Module is used to reconstruct the LBD frames to HBD frames.

**Limitations:**

**The main motive of the paper is to analyze the generalization perspective of the BDE models. No study has shown why and how only these 14 classes are aiding in generalization.
The dataset used for training comes from these 14 degradations and no reference is provided if this is taken from some other paper (I think the first contribution of the paper is pointing to the 14 degradations only). Therefore, a clear study is needed on the robustness and generalization capability of 14 degradations.

**No mention of how the degradation model mentioned in this paper is achieving DQ, Quantization, and Color Conversion. No architecture is provided.
This is necessary because the encoding generated from the classification model, which is a key contribution of the paper, depends on how the degradation is achieved.

**No explanation is provided on how the calculated loss is back propagated in this complex architecture.

**The reference of the SA and CA block is missing (used in the ATA).

**The details about how the architecture is expanding bits are not provided. Also, no reference is provided to infer that.

**Suitability:**

3

---

### Meta-Review · Area_Chair_yjFg · 2024-07-08

**Recommendation:** Accept (Poster)
**Confidence:** 3

**Metareview:**

Based on the reviews, author rebuttal, and updates, the paper slightly leans toward acceptance but still could be improved to address remaining issues toward submission of camera-ready version of the paper. Thus, the recommendation is accept and shepherding is recommended (in general, this is recommended for all accepted papers).